# Decreased Activity of Circulating Butyrylcholinesterase in Blood Is an Independent Prognostic Marker in Pancreatic Cancer Patients

**DOI:** 10.3390/cancers12051154

**Published:** 2020-05-04

**Authors:** Eva Valentina Klocker, Dominik Andreas Barth, Jakob Michael Riedl, Felix Prinz, Joanna Szkandera, Konstantin Schlick, Peter Kornprat, Karoline Lackner, Jörg Lindenmann, Herbert Stöger, Michael Stotz, Armin Gerger, Martin Pichler

**Affiliations:** 1Division of Clinical Oncology, Department of Medicine, Comprehensive Cancer Center Graz, Medical University of Graz, 8010 Graz, Austria; eva.klocker@medunigraz.at (E.V.K.); dominik.barth@medunigraz.at (D.A.B.); j.riedl@medunigraz.at (J.M.R.); felix.prinz@medunigraz.at (F.P.); joanna.szkandera@medunigraz.at (J.S.); herbert.stoeger@medunigraz.at (H.S.); michael.stotz@medunigraz.at (M.S.); armin.gerger@medunigraz.at (A.G.); 2Research Unit “Non-coding RNAs and Genome Editing in Cancer”, Medical University of Graz, 8010 Graz, Austria; 3Department of Experimental Therapeutics, The University of Texas MD Anderson Cancer Center, Houston, TX 77030, USA; 43rd Medical Department with Hematology and Medical Oncology, Hemostaseology, Rheumatology and Infectious Diseases, Laboratory for Immunological and Molecular Cancer Research, Oncologic Center, Paracelsus Medical University Salzburg, 5020 Salzburg, Austria; k.schlick@salk.at; 5Division of General Surgery, Department of Surgery, Medical University of Graz, 8010 Graz, Austria; peter.kornprat@medunigraz.at; 6Institute of Pathology, Medical University of Graz, 8036 Graz, Austria; karoline.lackner@medunigraz.at; 7Division of Thoracic Surgery, Department of Surgery, Medical University of Graz, 8010 Graz, Austria; jo.lindenmann@medunigraz.at

**Keywords:** butyrylcholinesterase, prognostic factor, pancreatic cancer

## Abstract

*Introduction:* The activity of butyrylcholinesterase (BChE) in blood reflects liver function and has recently been associated with systemic inflammatory response and tumor cachexia. As these conditions have been previously linked with pancreatic cancer (PC), the purpose of the present study was to evaluate the prognostic impact of plasma BChE in PC. *Methods:* Data from 574 consecutive PC patients, treated between 2004 and 2018 at a single academic center, was evaluated. The primary endpoint was cancer-specific survival (CSS), analyzed by Kaplan–Meier curve, and both univariate and multivariate Cox proportional models. *Results:* BChE activity negatively correlated with other liver parameters (bilirubin, gamma-glutamyl transferase (GGT), aspartate aminotransferase (AST), alkaline phosphatase (ALP), and C-reactive protein (CRP)), and positively correlated with albumin levels, respectively (*p* < 0.01). In univariate analysis, a low plasma BChE activity was a factor of poor CSS (hazard ratio: 1.4, 95% confidence interval: 1.129–1.754, *p* = 0.002). In multivariate analysis, tumor stage, tumor grade, administration of chemotherapy, bilirubin levels and a low BChE activity (hazard ratio: 1.42, 95% confidence interval: 1.10–1.82; *p* = 0.006) were identified as independent prognostic factors. *Conclusion*: Decreased activity of BChE in blood plasma predicts shorter survival time in PC patients. Therefore, BChE might be helpful in additional stratification of patients into different prognostic risk groups.

## 1. Introduction

Pancreatic cancer (PC) ranges among the most aggressive tumor types with a five-year survival rate varying between two to nine percent worldwide [1]. In the United States, the five-year-survival rate determined for the period between 2008 and 2014 was 34 percent when diagnosed at early local tumor stages, while it was reduced to 12 percent in locally advanced stages. When distant metastases are already present, the five-year-survival is only three percent [2]. In 2019, PC was the fourth leading cause of cancer related mortality despite the fact that PC was the 10^th^ most common cancer type in men and the 9th most common in women [2]. Typically, at the time of diagnosis only a small number of patients are identified at a stage that allows curative resection. Prognostic biomarkers are helpful in identifying patients with poor clinical outcome, and may help to stratify patients into clinical trials and more intense treatment plans [3].

Currently applied prognostic variables include histological grading, assessment of lymph node metastases, measurement of tumor size and determination of intrapancreatic perineural invasion [4,5,6,7]. However, the majority of these established clinico-pathological prognosticators are only available for assessment after surgery, making treatment decisions based on prognostic markers in metastatic PC more difficult. Measurement of novel molecular biomarkers are associated with high costs, time-consuming procedures and laboratory efforts [8,9]. Therefore, the search and establishment of easily determinable and available pre-treatment prognostic biomarkers is warranted and intensively studied [10,11,12]. 

Cholinesterases are enzymes that catalyze the hydrolysation of acetylcholine and other choline esters with high activity. There are two types of cholinesterases: Firstly, the acetylcholinesterase (AChE), which is mainly located in the nervous system, muscles and in erythrocytes and shows particularly high affinity to acetylcholine [13]. Secondly, the butyrylcholinesterase (BChE), also called pseudocholinesterase, plasma cholinesterase or serum cholinesterase, which is an α-glycoprotein with lower affinity to acetylcholine and which is located in the nervous system, liver and various other tissues [13]. Circulating BChE activity is an established marker for measuring liver synthesis and the nutritional status [14,15]. Moreover, there is evidence for its indicative role in systemic inflammation [16,17].

As PC frequently affects the liver´s functional state and nutritional status through the formation of liver metastases, portal vein thrombosis and systemic inflammatory responses, alterations in the BChE activity may reflect the aggressiveness of the disease. Therefore, we aimed at assessing the potentially prognostic value of BChE together with several other liver parameters in PC patients.

## 2. Results

The patient cohort in this study comprised a total of 574 patients, with 268 (46.7%) females and 306 (53.3%) males, all of which had histologically confirmed pancreatic adenocarcinoma. Median age in the cohort was 66 years (interquartile range: 58–72 years). Of this cohort, 25% had stage I or II disease, 5.4% had stage III disease and 69.6% of the patients, representing the largest sub-group, were diagnosed with stage IV metastatic disease, respectively. The tumor grading was G1 or G2 in 59.8% of patients, and G3 or G4 in 40.2%. In our cohort, 411 patients (71.4%) received chemotherapy and 173 (30.1%) of all 574 patients underwent surgical resection. Overall, the median survival was seven months (range 0–112 months) in our cohort. A summary of these patient characteristics is delineated in Table 1.

The median value of the liver parameters determined in the patient cohort was the following: total serum bilirubin, 0.8 mg/dL (normal range: 0.10–1.2 mg/dL; interquartile range: 0.48–2.07); GGT (gamma-glutamyl transferase), 157.50 U/L (upper normal limit: 55 U/L; interquartile range: 47.75–429.00); AST (aspartate aminotransferase), 34 U/L (upper normal limit: 45 U/L; interquartile range: 23.00–68.00); ALT (alanine aminotransferase), 45 U/L (upper normal limit: 35 U/L; interquartile range: 24.00–103); ALP (alkaline phosphatase), 145.5 U/L (normal range: 40–130 U/L; interquartile range: 84.25–145.00); PT (prothrombin time), 98% (normal range: 70–120%; interquartile range: 88–105.5); BChE, 6565 U/L (normal range: 3900–11,000 U/L; interquartile range: 5457–7937); CA 19-9, 809.5 U/L (upper normal limit: 37 U/L; interquartile range: 116–6237).

By applying ROC (receiver operating curve) analysis, the optimal cut-off to differentiate between survival in our patient cohort for BChE was identified to be ≤7272 U/L. Dividing the cohort into two groups according to the cut-off value we observed a significant association between low BChE activity and poor patient survival (Figure 1). The median survival time was 11 months (95% confidence interval: 9.44–12.56) vs. 7 months (95% confidence interval: 5.95–8.05) in the high vs. low BChE group, respectively (*p*-value = 0.001, log-rank test).

The BChE activity was significantly negatively correlated with bilirubin levels (R = −0.295), GGT (R= −0.195), AST levels (R = −0.126), ALP (R = −0.221), C-reactive protein (CRP) levels (R = −0.361) and age (R = −0.134), whereas it was positively correlated with albumin levels (R = 0.499) and PT (R = 0.125) (*p*-values < 0.01 for all tested variables, Spearman correlation). No correlations were observed for CEA levels, CA 19-9 and neutrophil-lymphocyte ratio (*p* > 0.05). A significant correlation was identified for body mass index (BMI) (R = 0.113, *p* = 0.023). No significant association of BChE activity and tumor grade, stage or Karnofsky performance status was observed (*p* > 0.05).

Univariate and multivariate Cox analyses were also applied to examine the prognostic value of BChE in relation to other clinico-pathological parameters. Univariate analysis demonstrated the prognostic value of surgical resection (hazard ratio:0.339, 95% confidence interval: 0.775–0.418, *p* < 0.001), tumor grading (hazard ratio:1.269, 95% confidence interval: 1.065–1.512, *p* = 0.008), high tumor stage (hazard ratio: 3.789, 95% confidence interval: 2.995–4.794, *p* < 0.001), chemotherapy (hazard ratio: 0.412, 95% confidence interval: 0.339–0.501, *p* < 0.001) and CA 19-9 levels (hazard ratio: 1.872, 95% confidence interval: 1.554–2.256, *p* < 0.001).

By using the calculated optimized cut off values for all laboratory variables, we found bilirubin (hazard ratio: 0.746, 95% confidence interval: 0.610–0.913, *p* = 0.004), GGT (hazard ratio: 1.443, 95% confidence interval: 1.093–1.905, *p* = 0.010), ALT (hazard ratio: 0.791, 95% confidence interval: 0.658–0.951, *p* = 0.013), ALP (hazard ratio: 1.440, 95% confidence interval: 1.101–1.884, *p* = 0.008) and BChE (hazard ratio: 1.406, 95% confidence interval: 1.129–1.754, *p* = 0.002) to show a significant association with CSS (cancer-specific survival) in univariate analysis (Table 2).

In multivariate analysis only bilirubin (hazard ratio: 0.694, 95% confidence interval: 0.502–0.96, *p* = 0.027) and BChE (hazard ratio: 1.416, 95% confidence interval: 1.10–1.181, *p* = 0.006) remained as independent prognostic markers (Table 2). In addition, CA 19-9 remained as a significant predictor of CSS in multivariate analysis (hazard ratio: 1.288, 95% confidence interval: 1.015–1.635, *p* = 0.037) (Table 2). 

Furthermore, as shown in Table 2, our analysis demonstrates that tumor grading (hazard ratio: 1.699, 95% confidence interval: 1.342–2.15, *p* < 0.001), high tumor stage (hazard ratio: 3.001, 95% confidence interval: 2.178–4.136, *p* < 0.001) and chemotherapy (hazard ratio: 0.329, 95% confidence interval: 0.251–0.432), *p* < 0.001) were significant independent prognostic markers of CSS when analyzed by multivariate Cox proportional analysis (Table 2).

## 3. Discussion

In our study, we identified for the first time an association between low activity of BChE in plasma samples at time of diagnosis and poor CSS in a large cohort of PC patients. In general, BChE is well known as a marker for liver function and serves as an indicator of the nutritional status evaluated in daily routine [14,15]. Given the fact that PC tends to frequently metastasize in the liver, low BChE activity may reflect the decreased liver function due to the presence of multiple metastases. In our study, we observed a significant correlation of BChE and BMI, which might also reflect the association with the nutritional status. In addition, bilirubin appeared to carry potential as an additional independent prognostic marker in multivariate analysis. Bilirubin was also positively correlated to the localization of tumors in the pancreatic head, which may indicate that these cancers might have been diagnosed in earlier disease stages due to jaundice provoked by cholestasis, and may therefore have a better prognosis. In contrast, our study did not reveal an association between GGT, ALT, AST and prothrombin time in CSS in multivariate analysis, although these parameters were negatively correlated with the BChE activity.

To the best of our knowledge, this is the largest study investigating the usefulness of BChE as a prognostic biomarker in a cohort of PC patients. In a previously published study including 75 PC patients at disease recurrence after local therapy, BChE activity below 300 U/L was identified as an independent prognostic marker in multivariate analysis in patients without peritoneal dissemination. Nevertheless, there was no association with the presence of liver metastases. Furthermore, low BChE activity was associated with histologically confirmed nerve plexus invasion at the time of curative resection as well with anemia, poor performance status, cachexia, hypoalbuminemia, hypocholesterinemia and ascites, all signs of exceedingly advanced disease [18]. However, this retrospective study only included PC patients at recurrence following curative resection, whereas our study explored a cohort consisting of PC patients at primary diagnosis across all tumor stages, establishing the BChE activity as a prognostic marker in a broader patient spectrum. This data is in line with reports for other types of tumors. For instance, low BChE level was revealed as an independent marker of shorter survival time in colorectal carcinoma, upper tract urothelial carcinoma, clear cell renal cell carcinoma, prostate cancer and cervical cancer [19,20,21,22,23,24]. In gastric cancer patients, a lower activity of BChE when compared to healthy controls has been described [25]. Santarpia et al. found lower levels of BChE in terminal cancer patients receiving parenteral nutrition who had shorter survival time [26]. However, a study by Prabhu et al. including a small cohort of oral squamous cell carcinoma patients (*n* = 39) showed an increase in BChE levels in cancer patients compared to healthy controls [27]. In a retrospective study by Pavo et al. aiming to find independent pre-treatment prognostic liver parameters in various cancer entities for the prediction of all-cause mortality, both low levels of BChE and low levels of albumin were identified as independent prognostic parameters. They were independent of primary and secondary hepatic involvement at the time of diagnosis [28]. Interestingly, different liver parameters such as bilirubin, ALT, AST and GGT did not turn out as prognostic parameters in this analysis, a similar finding to our study [28].

Similar to our findings, a study by Lampón et al. found a negative correlation between CRP levels and BChE activity in patients with chronic systemic inflammation [16]. In their study, they suggest that BChE may be a negative inflammatory reactant [16]. Acetylcholine suppresses the production of pro-inflammatory cytokines such as TNF, IL-1 β, IL-6 and IL-18 through the cholinergic anti-inflammatory pathway and as a result regulates immune reactions [29,30]. It is supposed that an enhanced activity of cholinesterases such as AChE and BChE play a role in systemic inflammation through suppression of the cholinergic anti-inflammatory pathway by hydrolytic destruction of acetylcholine [17]. Furthermore, Pavo et al. showed a significant inverse correlation between BChE levels and inflammatory markers such as CRP, IL-6 and serum amyloid A in cancer patients [28]. Another study showed a significant association between cholinesterase activities including BChE with IL-6 and TNF-α but not with CRP in frail elderly patients without cancer [31]. Possibly, low BChE activity may be an indicator of systemic inflammatory reaction.

Our study has some limitations, mainly due to its retrospective nature. BChE activity is known to be a marker of impaired liver synthesis, meaning that other non-cancer related pre-existing liver diseases might influence the activity and contribute as co-morbidities and confounders to the prognostic significance of the marker. Furthermore, we did not determine the genetic variants of BChE in single patients before blood sampling, although these genetic variants can bear different activities of BChE in plasma [32,33].

Nevertheless, to the best of our knowledge, our study represents to date the largest one validating the prognostic value of BChE activity in PC patients.

## 4. Materials and Methods

In our study, we included data of 574 patients with a histologically verified adenocarcinoma of the pancreas. All patients were treated at the Division of Clinical Oncology, Medical University of Graz, between 2004 and 2018. Data regarding clinico-pathological variables and laboratory values were retrieved from medical records at the Division of Clinical Oncology and pathological records from the Institute of Pathology. Staging was performed in accordance with the 7th edition TNM classification system [34]. For analysis we evaluated selected laboratory values obtained within 7 days up to two weeks before the date of diagnosis or treatment, including the following parameters: bilirubin, ALP, GGT, ALT, AST, PT and BChE. Liver enzymes and BChE activity were measured using a Cobas automatic biochemical analyzer with matched reagent kits (Roche, Basel, Switzerland). Plasma BChE activity was determined using Cobas cholinesterase kit and butyrylthiocholine served as the substrate. Patient clinical and radiological conditions were evaluated every three months during the first three years, every six months over the following five years, and finally every year for curative resected tumor stages. Date of death was retrieved from the central registry of the Austrian Bureau of Statistics. There were no drop-outs due to a lack in proper follow-up. Our study was approved by the local ethics committee of the Medical University of Graz (Ethics Committee of the Medical University of Graz, Austria; document number No. 26-196 ex 13/14). Because of the retrospective data collection, there was no requirement for a written informed consent by the individual patients. Instead, we had a “waiver of consent” granted by the local ethics committee.

### Statistical Analyses

We determined CSS as the time (in months) between date of diagnosis and cancer-related death. Statistical analyses were performed using Statistical Package for Social Sciences version 20.0 (SPSS Inc., Chicago, IL, USA) and MedCalc (Windows version 18.5, MedCalc Software bvba, Ostend, Belgium). For analysis of the relation between clinico-pathological parameters and plasma laboratory values we utilized non-parametric tests (Mann-Whitney U and χ^2^ test). To define an optimized cut-off value, ROC analysis was performed [35,36]. Kaplan–Meier analysis was applied to estimate patient survival status in test versus comparison group, which was done by using log-rank test. To define independent clinico-pathological factors influencing CCS we used backward stepwise multivariate Cox analysis. Estimated hazard ratios derived from Cox analyses were defined as relative risks with 95% confidence intervals. We considered a two-sided *p* < 0.05 as statistically significant.

## 5. Conclusions

In conclusion, in the present study, lower activity of BChE in plasma was demonstrated to represent a prognostic factor in PC patients. This simple, highly reproducible, inexpensive and easily available marker shows a potential to select patients at high risk for poor clinical outcome for appropriate treatment strategies.

## Figures and Tables

**Figure 1 cancers-12-01154-f001:**
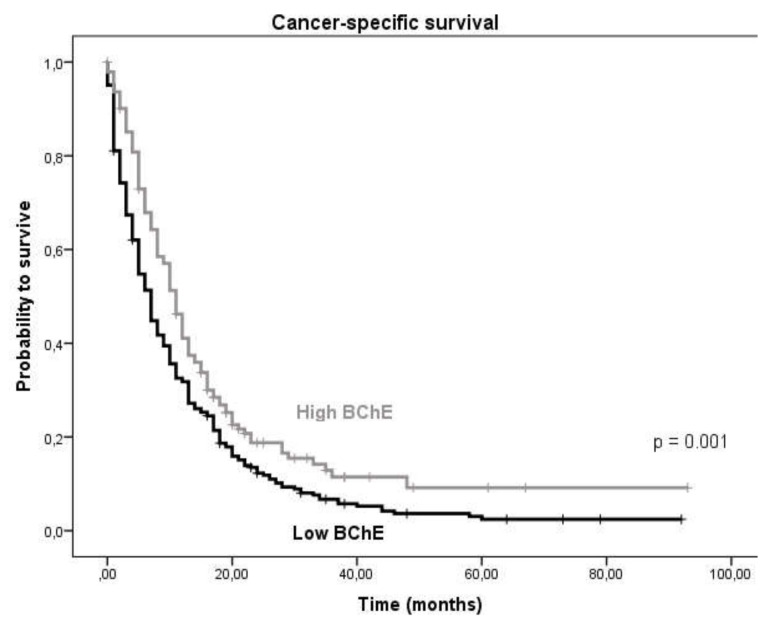
CSS (cancer-specific survival) in patients with low BChE activity vs. high BChE activity.

**Table 1 cancers-12-01154-t001:** Clinico-pathological characteristics of patients comprising the study cohort (*n* = 574).

Characteristics	No. Pancreatic Cancer (%)
**Gender** female male	268 (46.7) 306 (53.3)
**Tumor stage** I + II III IV	143 (25) 31 (5.4) 399 (69.6)
**Tumor grade** 1 + 2 3 + 4	343 (59.8) 231 (40.2)
**Surgical resection** yes no	173 (30.1) 401 (69.9)
**Chemotherapy** missing cases yes no	1 (0.2) 410 (71.4) 163 (28.4)
**Karfnosky Index** missing cases ≤80 90–100	5 (9) 337 (58.7) 232 (40.4)
**Cancer specific survival** alive dead	53 (9.2) 521 (90.8)

**Table 2 cancers-12-01154-t002:** Univariate and multivariate Cox proportional analysis regarding CSS in pancreatic cancer patients.

Variable	Subset	Univariate Analysis		Multivariate Analysis	
		HR ^1^ (95% CIl) ^2^	*p*	HR ^1^ (95% CI) ^2^	*p*
**Gender**	Female/Male	1.159 (0.975–1.377)	0.94	1.003 (0.794–1.266)	0.98
**Grading**	G3+4/G1+2	1.269 (1.065–1.512)	0.008	1.699 (1.342–2.15)	<0.001
**Staging**	Stage III/I+II	3.161 (2.099–4.761)	<0.001	2.254 (1.367–3.717)	0.001
	Stage IV/I+II	3.789 (2.995–4.794)	<0.001	3.001 (2.178–4.136)	<0.001
**Chemotherapy**	Yes/No	0.412 (0.339–0.501)	<0.001	0.329 (0.251–0.432)	<0.001
**Surgical resection**	Yes/No	0.339 (0.775–0.418)	<0.001	not included	
**CA 19-9**	>1191.7/≤1191.7 U/mL	1.872 (1.554–2.256)	<0.001	1.288 (1.015–1.635)	0.037
**Bilirubin**	>1.9/≤1.9 mg/dL	0.746 (0.610–0.913)	0.004	0.694 (0.502–0.96)	0.027
**GGT**	>25/≤25 U/L	1.443 (1.093–1.905)	0.010	1.1 (0.711.686)	0.663
**AST**	>42/≤42 U/L	0.880 (0.737–1.052)	0.160	1.017 (0.713–1.45)	0.925
**ALT**	>64/≤64 U/L	0.791 (0.658–0.951)	0.013	0.876 (0.623–1.231)	0.446
**ALP**	>70/≤70 U/L	1.440 (1.101–1.884)	0.008	1.406 (0.937.11)	0.1
**BChE**	≤7272/>7272 U/L	1.406 (1.129–1.754)	0.002	1.416 (1.10–1.818)	0.006
**PT**	>70/≤70%	0.706 (0.485–1.028)	0.069	0.777 (0.466–1.293)	0.331

Bold values indicate significance (*p* ≤ 0.05). ^1^ HR, hazard ratio; ^2^ CI, confidence interval.

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
