# Peer review of "Decreased Activity of Circulating Butyrylcholinesterase in Blood Is an Independent Prognostic Marker in Pancreatic Cancer Patients"

_cancers, 2020, doi:10.3390/cancers12051154_

Round 1

Reviewer 1 Report

THe study by M Pichler and coworkers is a statistical analysis of the pertinence of using BChE levels as prognostic marker in pancreatic cancer. Th whole paper is celarly writen, and the statistical analysis relevent, and the conclusion is clearly supported by the study. The only point that should be corrected is the lack of references to the analytical method used to determine the enzyme levels, and a comparison between BChE and AChE levels in patients sera. Overall, due to the quality of this paper, it is consistent with publication in "Cancer".

Author Response

Reviewer #1: THe study by M Pichler and coworkers is a statistical analysis of the pertinence of using BChE levels as prognostic marker in pancreatic cancer. Th whole paper is celarly writen, and the statistical analysis relevent, and the conclusion is clearly supported by the study.

Authors reply: We thank the for appreciating our work and his supporting words.

Reviewer #1: The only point that should be corrected is the lack of references to the analytical method used to determine the enzyme levels, and a comparison between BChE and AChE levels in patients sera. Overall, due to the quality of this paper, it is consistent with publication in "Cancer".

Authors reply: We have now added this information in the methods part of the manuscript.

Reviewer 2 Report

Summary: A retrospective study of laboratory data for 574 patients with histologically confirmed pancreatic adenocarcinoma identified plasma butyrylcholinesterase activity at the time of diagnosis as a good predictor of survival time. Patients with a butyrylcholinesterase activity of ≤7272 U/L were classified in the low activity group, while those with activity higher than 7272 U/L were classified in the high activity group. Patients in the low butyrylcholinesterase activity group survived a median of 7 months. Those in the high activity group survived a median of 11 months.

Minor comments

  1. Language problem: the term “poor prognostic marker “ is used in the title, abstract, and on line 164. This wording means that butyrylcholinesterase activity is not a good predictor of survival for pancreatic cancer patients. This is the opposite of what the authors mean to say. It is suggested to delete the word “poor” from the title.

            The last sentence in the abstract could be rewritten “Decreased activity of BChE in blood plasma predicts shorter survival time in PC patients…..”

            Line 164 could be rewritten “low BChE level was revealed as an independent marker of shorter survival time in colorectal carcinoma…….”

  1. Please add a reference for the butyrylcholinesterase activity assay used in the Division of Clinical Oncology.

Author Response

Reviewer #2:

Summary: A retrospective study of laboratory data for 574 patients with histologically confirmed pancreatic adenocarcinoma identified plasma butyrylcholinesterase activity at the time of diagnosis as a good predictor of survival time. Patients with a butyrylcholinesterase activity of ≤7272 U/L were classified in the low activity group, while those with activity higher than 7272 U/L were classified in the high activity group. Patients in the low butyrylcholinesterase activity group survived a median of 7 months. Those in the high activity group survived a median of 11 months.

Minor comments

Reviewer #2: Language problem: the term “poor prognostic marker “ is used in the title, abstract, and on line 164. This wording means that butyrylcholinesterase activity is not a good predictor of survival for pancreatic cancer patients. This is the opposite of what the authors mean to say. It is suggested to delete the word “poor” from the title.

Authors reply: We apologize for this misleading wording and have now adapted the term accordingly.

Reviewer #2: The last sentence in the abstract could be rewritten “Decreased activity of BChE in blood plasma predicts shorter survival time in PC patients…..”

Authors reply: We have adapted the statement accordingly.

Reviewer #2: Line 164 could be rewritten “low BChE level was revealed as an independent marker of shorter survival time in colorectal carcinoma…….”

Authors reply: We have substituted the sentence accordingly.

Reviewer #2: Please add a reference for the butyrylcholinesterase activity assay used in the Division of Clinical Oncology.

Authors reply: We have now added a reference for the BCHe assay accordingly.